# The Role of Chemokines in the Pathophysiology of Major Depressive Disorder

**DOI:** 10.3390/ijms20092283

**Published:** 2019-05-09

**Authors:** Vladimir M. Milenkovic, Evan H. Stanton, Caroline Nothdurfter, Rainer Rupprecht, Christian H. Wetzel

**Affiliations:** Department of Psychiatry and Psychotherapy, Molecular Neurosciences, University of Regensburg, D-93053 Regensburg, Germany; evan.h.stanton@gmail.com (E.H.S.); Caroline.Nothdurfter@medbo.de (C.N.); Rainer.Rupprecht@medbo.de (R.R.); Christian.Wetzel@klinik.uni-regensburg.de (C.H.W.)

**Keywords:** major depressive disorder, chemokines, neuroinflammation

## Abstract

Major depressive disorder (MDD) is a debilitating condition, whose high prevalence and multisymptomatic nature set its standing as a leading contributor to global disability. To better understand this psychiatric disease, various pathophysiological mechanisms have been proposed, including changes in monoaminergic neurotransmission, imbalance of excitatory and inhibitory signaling in the brain, hyperactivity of the hypothalamic-pituitary-adrenal (HPA) axis, and abnormalities in normal neurogenesis. While previous findings led to a deeper understanding of the disease, the pathogenesis of MDD has not yet been elucidated. Accumulating evidence has confirmed the association between chronic inflammation and MDD, which is manifested by increased levels of the C-reactive protein, as well as pro-inflammatory cytokines, such as Interleukin 1 beta, Interleukin 6, and the Tumor necrosis factor alpha. Furthermore, recent findings have implicated a related family of cytokines with chemotactic properties, known collectively as chemokines, in many neuroimmune processes relevant to psychiatric disorders. Chemokines are small (8–12 kDa) chemotactic cytokines, which are known to play roles in direct chemotaxis induction, leukocyte and macrophage migration, and inflammatory response propagation. The inflammatory chemokines possess the ability to induce migration of immune cells to the infection site, whereas their homeostatic chemokine counterparts are responsible for recruiting cells for their repair and maintenance. To further support the role of chemokines as central elements to healthy bodily function, recent studies suggest that these proteins demonstrate novel, brain-specific mechanisms including the modulation of neuroendocrine functions, chemotaxis, cell adhesion, and neuroinflammation. Elevated levels of chemokines in patient-derived serum have been detected in individuals diagnosed with major depressive disorder, bipolar disorder, and schizophrenia. Furthermore, despite the considerable heterogeneity of experimental samples and methodologies, existing biomarker studies have clearly demonstrated the important role of chemokines in the pathophysiology of psychiatric disorders. The purpose of this review is to summarize the data from contemporary experimental and clinical studies, and to evaluate available evidence for the role of chemokines in the central nervous system (CNS) under physiological and pathophysiological conditions. In light of recent results, chemokines could be considered as possible peripheral markers of psychiatric disorders, and/or targets for treating depressive disorders.

## 1. Introduction

Major depressive disorder (MDD) is a highly prevalent condition, and is the third leading cause of disability worldwide [1]. Despite the availability of numerous anti-depressive treatments, 30% of patients diagnosed with MDD fail to respond to anti-depressant therapy, or show only a partial response [2,3]. Bipolar disorder, which is characterized by recurrent depressive and manic episodes, is difficult to diagnose [4], and is often misdiagnosed as MDD, particularly during a depressive episode [5]. Diagnostic and Statistical Manual of Mental Disorders (DSM-5) criteria for unipolar and bipolar depression are the same during a major depressive episode [6]. Therefore, there is a need for novel biomarkers, which could distinguish between these two conditions [5]. This inadequate response to treatment reflects an incomplete understanding of the actual pathogenesis of depression, which was initially linked to changes in monoaminergic transmission [7,8]. Subsequent hypotheses include the disturbance of excitatory and inhibitory signaling in the brain [9,10], hyperactivity of the hypothalamic-pituitary-adrenal (HPA) axis [11,12], and hindrance upon the healthy progression of neurogenesis [13,14]. However, increasingly compelling lines of evidence indicate a role of nearly or completely asymptomatic subclinical systemic inflammation in the pathophysiology of MDD [15,16,17,18,19,20,21,22,23,24,25,26]. While using the reassessment of immune privilege in the central nervous system [27,28] as a foundation, complex interactions between the immune system and the brain began to emerge. The immune system regulates key aspects of brain development, neurogenesis, central nervous system (CNS) homeostasis, mood, and behavior [29,30,31,32,33,34,35]. As such, perturbations of the neuroimmune functions have been implicated in a number of psychiatric disorders, including MDD [36,37,38,39], bipolar disorder [40,41], schizophrenia [42,43,44,45], and autism [46,47].

Recent advances in neuroscience have linked chemotactic cytokines (chemokines) to neurobiological processes relevant to psychiatric disorders, such as synaptic transmission and plasticity, neurogenesis, and neuron-glia communication [48,49,50,51]. The disruption of any of these functions, by activation of the inflammatory response system, could be central for the pathogenesis of MDD. Impaired CXCL12/CXCR4 signaling is implicated in abnormal development, proliferation, and migration of neural progenitor cells [52,53], which is suggestive of their essential roles in mammalian neurogenesis. Furthermore, the dysregulation of various chemokines, which modulate neuronal activity by means of inducing signal transduction [54,55] and Ca^2+^ mobilization [56,57], could also be involved in pathophysiological processes leading to MDD. To add to the wide breadth of chemokine functionality, these ligands and their receptors, which are widely expressed in the CNS [58,59,60,61,62], coordinate immune cell recruitment and their subsequent migration to sites of inflammation. Therefore, this links peripheral and central inflammation. This phenomenon can be observed in the quantitative increase of chemokine concentrations within the serum of patients with MDD, relative to homeostatic levels. Moreover, this discrepancy is associated with the onset and progression of depression in humans [63].

To further investigate the potential connection between chemokines and depression, chemokine receptor knockout mice (CCR6 and CCR7) were created and observed to display behavioral phenotypes similar to psychiatric disorders, including MDD [64].

Altogether, these data provide evidence of the involvement of chemokines in processes underlying MDD. In this work, we will examine the role of chemokines in healthy and depressed states, as well as summarize to the best of our knowledge evidence to date for the possible role of chemokines in the pathogenesis of MDD.

## 2. Chemokine Superfamily

The chemokine superfamily contains a large number of ligands and receptors, which are classified into four sub-families (CXC, CC, C, and CX3C) [65], according to the number and spacing of their two N-terminal, disulfide bonding participating cysteine residues. Chemokines are small (8–12 kDa) heparin binding proteins, structurally related to cytokines that can induce directed chemotaxis of immune cells. However, chemokines are additionally involved in the regulation of migration of immune cells [66,67], blood-brain barrier (BBB) permeability [68], and synaptic pruning processes [69]. In addition to their structural criteria, chemokines can be subdivided into inflammatory chemokines, which are upregulated under inflammatory conditions, homeostatic chemokines that are responsible for maintaining homeostasis, and chemokines, which exhibit dual functionality [70].

The chemokine superfamily has expanded rapidly after the initial identification of secreted platelet factor 4 (PF4/CXCL4) [71] in 1977. Subsequent studies have identified more than 50 chemokines, as well as 20 chemokine receptors [72]. The majority of human chemokine genes are clustered on chromosomes 4 and 17. CXC chemokines can be found at chromosomal location 4q12-21, whereas most of the CC chemokines are located at 17q11-21 [73]. This suggests a rapid evolution by repeated gene duplications [74]. All chemokines share a very similar tertiary structure [75], including a highly flexible N-terminal domain and a long rigid loop, which are essential for interacting with their respective receptors [76], and a C-terminal α-helix. Typically, a given chemokine can bind to more than one receptor (Table 1) and, correspondingly, a number of different chemokines can be recognized by the same receptor [65]. Chemokines are secreted in response to inflammatory cytokines, and they selectively recruit monocytes, lymphocytes, and neutrophil-inducing chemotaxis by activating G-protein-coupled receptors (GPCRs) [77].

## 3. Chemokines and Chemokine Receptors in the Brain

Chemokines and their receptors are broadly expressed in the CNS in both physiological and pathophysiological states [58,59,60,78]. The glia cells (astrocytes, oligodendrocytes, and microglia), and neuronal cells constitutively express several chemokines, including CCL2, CCL3, CCL19, CCL21, CXCL10, and CX3CL1 [58,78,79,80], as well as others, which can be upregulated in response to pathological conditions. Endothelial cells of the BBB may, under severe inflammatory conditions, likewise produce several chemokines such as CCL2 [68], CCL4 and CCL5 [81], which bind CCR1, CCR2, and CCR5 [82] chemokine receptors that are expressed by circulating mononuclear cells.

In addition to their traditional role in immune surveillance and immune cell chemotaxis, chemokines and chemokine receptors residing in the brain are also involved in the homeostatic maintenance of the CNS through either autocrine or paracrine activity [83]. Different expression patterns of various chemokines during embryonic and postnatal development is suggestive of their essential role for typical brain development. For example, CXCL12 and its receptors CXCR4/CXCR7 are involved in the proliferation and migration of neural progenitor cells (NPC). They are distinctively expressed in both the developing and the adult brain [61,84]. On the other hand, the CX3CL1 chemokine (fractalkine) and its receptor CX3CR1, which are constitutively present in the CNS, act to modulate inflammatory responses of microglia by suppressing its neurotoxicity [85] by reducing levels of the tumor necrosis factor α (TNF-α) and nitric oxide (NO) [86]. Other chemokines such as CXCL1 and CXCL8 exert neuro-modulatory effect on the synapsis of cerebellar neurons [87].

Consequently, the chemokine system, which plays an important role in neurogenesis, neuron-glia communication, synaptic transmission, and plasticity under physiological and pathophysiological conditions, might participate in the pathogenesis of depression. Evidence in support of this claim is that alterations to all of the previously mentioned processes are consistently implicated in various psychiatric disorders including MDD [11,88].

## 4. Regulation of Neurogenesis and Neuronal Plasticity by Chemokines

The process of neurogenesis, by which new neurons are continuously generated in discrete brain regions of many vertebrate species including humans, is particularly prominent in the dentate gyrus of the hippocampus [89,90]. Initial studies in patients with recurrent major depression, which have shown stress-induced loss of the hippocampal volume, suggested association of hippocampal atrophy with depression [91,92]. Furthermore, the decrease in hippocampal volume was correlated with the total duration of the depressive episodes [93]. Further studies have established a link between reduced adult hippocampal neurogenesis with the pathophysiology of several psychiatric disorders, including anxiety and depression [78,90,94,95]. Therapeutic interventions, such as electro-convulsive and anti-depressive therapy [96,97], are, on the other hand, able to promote recovery from depression, in part by enhancing hippocampal neurogenesis.

Chemokines play an important role in the regulation of neuronal development and plasticity, proliferation, migration, and neural progenitor cell (NPC) differentiation [98,99]. Because of the significant redundancy in chemokine receptor-ligand interactions, most of the chemokine or chemokine receptor knockout animals are viable and show no apparent neural phenotype [100]. The only exception to this is the knockout mice from either CXCL12 or its receptor CXCR4, which are not viable and exhibit cerebellum malformation. This is suggestive of their essential role in the migration of the NPCs [101]. NPCs derived from the hippocampus and the subventricular zone (SVZ) express various chemokine receptors on their surface [102], which are important for the regulation of proliferation and differentiation of these cells. The CX3CL1 chemokine, which is abundantly present on mature neurons and astrocytes, and its receptor CX3CR1 that is mostly expressed on microglia cells [103], are additionally involved in the regulation of neurogenesis and neuroplasticity. The CX3CL1 chemokine regulates microglial synaptic pruning of mature neurons [104], modulates several neurotransmitter systems [105], and regulates the activation state of microglia [85]. Therefore, this influences the development and plasticity of the CNS. Exogenous application of the CX3CL1 chemokine further enhanced in vivo neurogenesis in aged rats by modulating the microglia phenotype [106]. Other chemokines such as CCL2, CCL21, and CXCL9, promote neuronal differentiation, whereas CCL2, CXCL1, and CXCL9 favor oligodendrocyte differentiation [107]. Further support for the association of adult hippocampal neurogenesis and MDD arise from the studies, which demonstrated that various chronic anti-depressive treatments stimulate hippocampal neurogenesis [108]. However, recent evidence suggests that the alterations in adult hippocampal neurogenesis are not solely responsible for the development of depression [109].

Altogether, chemokines play a significant role in both neurogenesis and neuronal plasticity, which are essential for proper brain functioning, and any disturbance in any of these functions could lead to a depressed state.

## 5. Chemokines and Neurotransmission in the Adult CNS

Chemokines and their respective receptors, which are constitutively expressed in glial cells and neurons [59,110,111,112], are responsible for homeostatic maintenance of the developed brain. Recent data suggest that chemokines present a unique class of neurotransmitters and neuromodulators that regulate cell survival and synaptic transmission [103,113]. For example, patch-clamp experiments performed in Purkinje neurons demonstrated an increase in spontaneous GABAergic activity upon the application of CXCL12 [56]. Application of CXCL12 in rat hypothalamic slices similarly caused an increase of GABA release from melanin-concentrating hormone neurons [114]. According to subcellular studies, chemokines are detected in presynaptic nerve terminals, where they co-localize with various neurotransmitters, and are released ensuing membrane depolarization [115,116,117]. CX3CL1, which co-localize with serotonin in neurons of the dorsal raphe nucleus, may indirectly inhibit serotonin neurotransmission by upregulating the sensitivity of serotonin dorsal raphe nucleus neurons to GABA inputs [50]. Furthermore, results from electrophysiological studies suggest that CCL2, CCL5, CCL22, CXCL12, CXCL8, and CX3CL1 chemokines can modulate the electrical activity in cortical, cerebral, hippocampal, and hypothalamic neurons [59,105,118,119,120,121].

Overall, the data presented in this case suggest a significant role of chemokines in neurotransmission and modulation of neurotransmitter release, which are increasingly being implicated in the pathogenesis of MDD.

## 6. Pre-Clinical Evidence Linking Changes in the Chemokine Network to Depressive Behavior

Animal models of psychiatric disease are a potent tool to investigate possible causes and treatments for human diseases. However, they face a number of challenges given the lack of objective diagnostic tests, biomarkers, and low predictive power [122]. Early animal-utilizing studies of depression investigated stress-response paradigms [123], and would often involve the subjugation of models to mild, unpredictable stressors that were either acute or chronic in application. The response, which is reasoned to be analogous to stress-induced depression in humans, involves the dysregulation of the hypothalamic-pituitary-adrenal (HPA) axis, as well as the neuroendocrine and neurotransmitter systems [124]. Findings indicate that immobilization and painful stress experiments demonstrated increased expression of CXCL1 chemokine in various regions of the CNS [125,126]. On the other hand, in a mouse model of depressive behavior based on chronic variable stress, no significant differences in expression of the CCL2 chemokine in hippocampus were found [127]. In prenatally stressed rats, as a further animal model of depressive behavior, levels of CCL2, and CXCL12 chemokines were upregulated in the hippocampus and prefrontal cortex, which is suggestive of excessive microglial activation [128]. Moreover, chronic anti-depressant treatment has been shown to revert those changes [129].

An alternative attempt to model depression-like behavior in animals involves inducing sickness-like behavior by administering inflammatory cytokines or lipopolysaccharide (LPS), which mimic the depressive symptoms induced by treatment of human patients with interferons [130]. CXCL1, CXCL10, and CCL5 were up-regulated in mice in which the depressive-like behavior was induced by application of Interferon α [131]. Rats treated with CXCL1 chemokines have shown a dose-dependent reduction in both spontaneous open field activity and burrowing behavior [132]. Peripheral administration of LPS have further induced the expression of CXCL1 and CCL2 in the prefrontal cortex, hypothalamus, and plasma of rats exposed to chronic, intermittent, cold stress [133]. Animals lacking CX3CR1 receptors experienced an increased duration of sickness-like behavior on the tail suspension test after peripheral LPS challenge [134], which additionally implicates the role of the chemokine system in sickness-like behavior. Virus-induced sickness-like behavior can additionally cause impaired learning and cognitive dysfunction by mechanisms that remain poorly understood. However, recent studies performed in mice have suggested a key role of an innate immune system of the brain in mediating the behavioral effects of viral infection [135,136]. Virus associated activation of a subpopulation of circulating monocytes expressing the CX3CR1 receptor causes release of TNF-α, which induces dendritic spine loss and motor learning impairment [135]. The exact mechanism by which monocytes modulate synaptic activity is not known, but there is evidence to suggest it is microglia-independent [137]. Brain endothelial cells, which serve as a natural barrier to interferon-induced sickness behavior, could also play an important role for the communication between the central nervous and immune systems [136].

Stress has been shown to play an important role in the etiology of neuropsychiatric diseases, including depression [138], and a number of animal studies have identified that exposure to stress greatly increases the risk of developing depression [139]. However, most of the stressors applied were artificial, and, thus, are not a representational model of stress exposure in humans, which is mostly social in nature [140]. Lately, alternative animal models of depression have begun to focus on psychosocial stress, particularly on a paradigm based on social defeat [141]. Repeated social defeat (RSD) in mice causes an exposure-dependent increase of CXCL1 and CXCL2 levels in the brain, which is indicative of higher leukocyte recruitment in the brain vasculature [142]. Animals repeatedly exposed to social defeat show decreased volume and cell proliferation in the hippocampus and prefrontal cortex, which can be reverted by an anti-depressant treatment [123], bearing similar resemblance to the human studies. Altogether, various animal models of depressive-like behavior have provided evidence for the involvement of a chemokine network in the pathophysiology of major depression.

## 7. Involvement of Chemokines in the Pathophysiology of MDD—Clinical Studies

Several studies in humans and animal models have linked elevated levels of chemokines with the depressive behavioral symptoms, particularly increased levels of circulating inflammatory chemokines. The majority of the published clinical studies included the CCL2 and CXCL8 chemokines, and were based on the detection of the chemokine expression in blood or cerebrospinal fluid [48,73]. CCL2, which belongs to the group of the inflammatory chemokines, has been implicated in the chemotactic migration of peripheral monocytes to the brain [143]. Significantly higher concentrations of CCL2 in the serum of depressed patients compared to controls were described in numerous studies [23,144,145]. Moreover, antidepressant drug treatment effectively reduced peripheral levels of the CCL2 chemokine [146]. Although a considerable number of publications, including recent meta-analyses [15,48,63,147], have reported an increased CCL2 expression in patients diagnosed with MDD, studies involving MDD patients with suicidal ideation have surprisingly shown unchanged or reduced levels of the chemokine [148,149]. Considering a similar correlative elevation of CCL2 levels reported in patients diagnosed with bipolar disorder [150], more research is needed in order to effectively use elevated serum CCL2 levels as a marker of MDD.

CXCL8 levels in blood samples from a total of 40 studies involving 3788 participants were significantly elevated in depressed subjects when compared to controls reported in a recent comprehensive meta-analysis performed by Leighton et al. [15]. However, these results were obtained only after exclusion of a subgroup with physical illness. Significant differences that were observed in CXCL8 chemokine levels only after restricting the analysis to healthy subjects, suggest that inflammatory changes of underlying physical disease could mask the changes in chemokine levels in depressed patients [15]. Plasma levels of CCL3, also known as macrophage inflammatory protein-1α (MIP-1α), were similarly increased in depressed patients compared to healthy control subjects [15,23,151,152]. A significant increase of blood levels were also shown for further chemokines including CCL11, CXCL4, and CXCL7 [15]. Inflammation can also play an important role in the etiology of bipolar disorder, which has been suggested by several studies [153,154,155], in which patients with bipolar disorder showed increased levels of CCL11 and CXCL10 in the plasma. On the other hand, plasma levels of another chemokine from the CC group, CCL4, decreased in depressed patients in several studies [15,148,156]. Many other chemokines examined, such as CCL5, CCL7, CXCL9, and CXCL10, showed no significant differences [15].

During depressive episodes, biochemical measurements indicate an increased level of the microglia-enriched protein, translocator protein 18 kDa (TSPO), which is elucidated by the correlative increase of binding by TSPO-specific ligands [157]. It is still a matter of debate whether an increased TSPO ligand binding in depression is due to the proliferation of microglial cells or infiltration of circulating macrophages, which also express high amounts of TSPO protein through the blood-brain-barrier (BBB). Our recent published data show higher levels of CCL22, macrophage-derived chemokine (MDC) in the blood of the MDD patients who responded to anti-depressive therapy [158]. Therefore, this suggests that chemotaxis and infiltration of monocytes, as well as recruiting T-helper 2 cells (Th2) and T-regulatory cells through the BBB, could play a significant role in the pathophysiology of MDD. A link between macrophages and depression was initially proposed in 1991 [159], where excessive activity of macrophages has been suggested as a key factor in the etiology of this illness. Recent studies in various models of CNS injury and neurodegenerative diseases have highlighted the essential role of infiltrating monocyte-derived macrophages for the CNS repair process by resolving inflammation [160].

Typical pharmacological treatment of MDD can also decrease peripheral inflammation, as demonstrated by the reduction in levels of CCL2 [146]. However, other approaches are necessary in order to improve treatment outcome. Targeting immune-related pathways, which are altered in MDD and in bipolar disorder, could constitute a novel therapeutic mechanism for the treatment of both MDD and BD [161,162]. CCL11, which has been associated with many psychiatric disorders and its CCR3 receptor, may have represented attractive targets for treating both MDD and BD [163]. Moreover, the use of nonsteroidal anti-inflammatory drugs, including celecoxib, as an adjunctive treatment in MDD patients, and minocycline demonstrate a significant anti-depressive effect [164,165]. Even electroconvulsive therapy, which is one of the most effective treatment options for treatment-resistant depression, modulates peripheral immune activation [166]. In order to provide an accurate diagnosis, and to monitor treatment response in MDD and BD patients, novel biomarkers are urgently needed. Biobanks with well-defined phenotype of MDD and BD patients [167,168,169] were established with a goal to expedite development of novel diagnostic and therapeutic compounds.

According to the available clinical studies reviewed in this work, it is clear that chemokines play an important role in regulating neurobiological processes relevant to psychiatric disorders, and that dysregulation of various chemokines could play an important role in the pathophysiology of MDD.

## 8. Conclusions

Elucidating the neurobiological basis of depression and the development of more effective pharmacological treatments are the principal challenges, and one of the main goals of modern medicine. Less than a third of MDD patients adequately respond to the initial antidepressant treatment, and over 35% of depressed patients fail to respond to different antidepressants altogether [170]. Considering that the majority of commonly prescribed anti-depressants act primarily by increasing or modulating monoamine neurotransmission [171], there is a need for novel therapeutic agents. Identification of specific biomarkers of depression, which could be used to predict a response to anti-depressive drugs, and develop new treatment options would help reduce the burden of depression.

An increasing body of evidence, reviewed in this study, demonstrates an important role for chemokines in the biology of depression. However, the majority of the studies were performed on peripheral blood samples, and had a cross-sectional design. In order to fully comprehend the changes that occur in depression, longitudinal studies with treated MDD patients will be necessary. An additional limitation of the majority of human studies published thus far is that the patho-physiological changes detected in the periphery might not reliably indicate changes in the CNS. Furthermore, many of the investigations utilized a small subset of chemokines, which limits our total understanding of inflammatory processes in vivo.

In summary, the data reviewed in this manuscript demonstrates the important role of chemokines in pathophysiology of MDD. Chemokines and their receptors, which are widely expressed in the CNS, could become novel diagnostic markers or therapeutic targets for MDD. However, additional research in larger populations, which should also include longitudinal studies, is necessary.

## 9. Methods

We performed literature searches through Pubmed and Google Scholar databases for articles published before September 2018. The search terms (chemokines OR cytokines OR neuroinflammation OR inflammation) AND (Depression OR Depressive Disorder OR Major Depressive Disorder) were used. Obtained references were additionally inspected and all relevant publications were included.

## Figures and Tables

**Table 1 ijms-20-02283-t001:** Chemokines and their known receptors. Chemokine receptors, which belong to the superfamily of GPCRs, can bind to multiple chemokines, and certain chemokines can similarly bind to more than one receptor. Adapted from Zlotnik and Yoshie 2012 [65].

Subfamily	Chemokine	Synonyms	Receptors
**CXC**	CXCL1	Growth-related oncogene α (GROα)	CXCR1/CXCR2
	CXCL2	Growth-related oncogene β (GROβ)	CXCR2
	CXCL3	Growth-related oncogene γ (GROγ)	CXCR2
	CXCL4	Platelet factor 4 (PF-4)	CXCR3-B
	CXCL5	Epithelial cell-derived neutrophil-activating factor 78 (ENA-78)	CXCR2
	CXCL6	Granulocyte chemoattractant protein (GCP-2)	CXCR1/CXCR2
	CXCL7	Neutrophil-activating protein (NAP-2)	CXCR1/CXCR2
	CXCL8	Interleukin-8 (IL-8)	CXCR1/CXCR2
	CXCL9	Monokine induced by γ-interferon (MIG)	CXCR3
	CXCL10	γ -interferon-inducible protein 10 (IP-10)	CXCR3
	CXCL11	Interferon-inducible T cell α -Chemoattractant (I-TAC)	CXCR3
	CXCL12	Stromal cell-derived factor 1 (SDF-1)	CXCR4
	CXCL13	B cell-activating chemokine 1 (BCA-1)	CXCR5
	CXCL14	Breast and kidney chemokine (BRAK)	CXCR4
	CXCL15	Lungkine	-
	CXCL16	Scavenger receptor for phosphatidylserine and oxidized lipoprotein (SR-POX)	CXCR6
	CXCL17	dendritic cell-attracting and monocyte-attracting chemokine-like protein (DMC)	CXCR8
**CC**	CCL1	I-309	CCR8
	CCL2	Monocyte chemoattractant protein 1 (MCP-1)	CCR2/CCR9/CCR11
	CCL3	Macrophage inflammatory protein 1α (MIP-1α)	CCR1/CCR5/CCR9
	CCL4	Macrophage inflammatory protein 1β (MIP-1β)	CCR1/CCR5/CCR9
	CCL5	Regulated on activation of normal T cell-expressed and secreted (RANTES) entities	CCR1/CCR3/CCR4/CCR5
	CCL7	Monocyte chemoattractant protein 3 (MCP-3)	CCR1/CCR2/CCR3
	CCL8	Monocyte chemoattractant protein 2 (MCP-2)	CCR2/CCR9/CCR11
	CCL11	Eosinophil chemotactic protein (Eotaxin-1)	CCR2/CCR3/CCR5
	CCL13	Monocyte chemoattractant protein 4 (MCP-4)	CCR2/CCR3/CCR5
	CCL14	Hemofiltrate CC chemokine (HCC1)	CCR1/CCR5
	CCL15	Leukotactin-1, macrophage inflammatory protein 5 (MIP-5)	-
	CCL16	Liver-expressed chemokine (LEC), monotactin-1 (MTN-1)	CCR1/CCR2/CCR5/CCR8
	CCL17	Thymus and activation-related chemokine (TARC)	CCR4
	CCL18	Macrophage inflammatory protein 4 (MIP-4)	CCR8
	CCL19	Epstein–Barr virus-induced receptor ligand chemokine (ELC)	CCR7
	CCL20	Liver-related and activation-related chemokine (LARC)	CCR6
	CCL21	Secondary lymphoid tissue chemokine (SCL)	CCR7
	CCL22	Macrophage-derived chemokine (MDC)	CCR4
	CCL23	Macrophage inflammatory protein 3 (MIP-3)	CCR1
	CCL24	Eosinophil chemotactic protein 2 (Eotaxin-2)	CCR3
	CCL25	Thymus lymphoma cell-stimulating factor (TECK)	CCR9
	CCL26	Macrophage inflammatory protein 4-α (MIP-4-α)	CCR3
	CCL27	Cutaneous T cell-attracting chemokine (CTACK)	CCR10
	CCL28	Mucosae-associated epithelial chemokine (MEC)	CCR10
**C**	XCL1	Lymphotactin-α	XCR1
	XCL2	Lymphotactin-β	XCR1
**CX3C**	CX3CL1	Fractalkine	CX3CR1

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
