# Peer review of "The Role of Chemokines in the Pathophysiology of Major Depressive Disorder"

_ijms, 2019, doi:10.3390/ijms20092283_

Round 1

Reviewer 1 Report

The authors conducted a review to summarize evidence supporting a putative role for chemokines in psychiatric disorders. The review is clear, well written and provides a nice overview of available clinical and experimental studies investigating the role of chemokines in neurodifferentiation, neurotransmission as well as in depression. 

I only suggest to add a short paragraph on the methodology used to conduct the bibliographic search.

Author Response

The authors would like to thank the reviewers for their constructive comments which helped to improve the manuscript.

Reviewer 1

"The authors conducted a review to summarize evidence supporting a putative role for chemokines in psychiatric disorders. The review is clear, well written and provides a nice overview of available clinical and experimental studies investigating the role of chemokines in neurodifferentiation, neurotransmission as well as in depression. 

I only suggest to add a short paragraph on the methodology used to conduct the bibliographic search."

In response to the reviewer’s suggestion, we have added Methods section to the manuscript, describing bibliographic search strategy.

Reviewer 2 Report

This is a nicely written and timely report.  One major deficiency is that authors did not discuss sufficiently possibilities that depression can be part of a different diagnosis and not only major depressive disorder.  This mainly relates to bipolar depression.  My recommendation is that they should address this and talk about how findings in major depressive disorder could also have implication for bipolar depression.  Addressing limitation on current diagnostic criteria based on DSM rather than neurobiology should also be discussed. In conclusion they could elaborate further on directions for future studies and development of biomarkers including offering some specifics.  For example,  mentioning existing biobanks may be helpful for readers to learn about existing resources.  Example for bipolar biobank is well described in

Int J Bipolar Disord. 2015 Dec;3(1):30. doi: 10.1186/s40345-015-0030-4. Epub  2015 Jun 24.

Development of a bipolar disorder biobank: differential phenotyping for subsequent biomarker analyses.  

They could also provide a similar example for major depressive disorder.

Author Response

The authors would like to thank the reviewers for their constructive comments which helped to improve the manuscript. We thank the reviewer for drawing our attention to this point. Even though emphasis of this manuscript was the role of chemokines in etiology of unipolar depression, we have now revised the manuscript and have addressed the individual points as follows:

"One major deficiency is that authors did not discuss sufficiently possibilities that depression can be part of a different diagnosis and not only major depressive disorder. This mainly relates to bipolar depression."

As the reviewer suggested, in introduction section of the manuscript, we briefly discuss the symptomatic differences between unipolar and bipolar depression.

"My recommendation is that they should address this and talk about how findings in major depressive disorder could also have implication for bipolar depression." 

As suggested by the reviewer, we have addressed potential impact of novel discoveries in MDD for bipolar depression in section 7 of the manuscript.

"Addressing limitation on current diagnostic criteria based on DSM rather than neurobiology should also be discussed."

In response to the reviewer’s suggestion, the shortcomings and limitations of the current diagnostic criteria based on DSM-V were briefly discussed.

"In conclusion they could elaborate further on directions for future studies and development of biomarkers including offering some specifics. For example, mentioning existing biobanks may be helpful for readers to learn about existing resources. "

We have now included future directions for development of novel biomarkers for bipolar disorder, and added references to biobanks for both major depressive and bipolar depressive disorders.

Round 2

Reviewer 2 Report

Authors addressed all comments appropriately.